# The Impacts of Regional Regulatory Policies on the Prevention and Control of Chronic Diseases in China: A Mediation Analysis

**DOI:** 10.3390/healthcare9081058

**Published:** 2021-08-18

**Authors:** Huihui Huangfu, Qinwen Yu, Peiwu Shi, Qunhong Shen, Zhaoyang Zhang, Zheng Chen, Chuan Pu, Lingzhong Xu, Zhi Hu, Anning Ma, Zhaohui Gong, Tianqiang Xu, Panshi Wang, Hua Wang, Chao Hao, Qingyu Zhou, Li Li, Chengyue Li, Mo Hao

**Affiliations:** 1Research Institute of Health Development Strategies, Fudan University, Shanghai 200032, China; 19111020043@fudan.edu.cn (H.H.); 19211020038@fudan.edu.cn (Q.Y.); zhouqingyu@fudan.edu.cn (Q.Z.); yiran_eric@126.com (L.L.); 2Collaborative Innovation Center of Social Risks Governance in Health, Fudan University, Shanghai 200032, China; pwshi@163.com (P.S.); shenqunhong108@163.com (Q.S.); zhangzhy@nhc.gov.cn (Z.Z.); chenzhengjd@163.com (Z.C.); puchuan68@sina.com (C.P.); lzxu@sdu.edu.cn (L.X.); aywghz@126.com (Z.H.); yxyman@126.com (A.M.); zhgong_zg@163.com (Z.G.); xtq1960@icloud.com (T.X.); wangpanshi03@163.com (P.W.); jswstwh@163.com (H.W.); 18906113216@189.cn (C.H.); 3Zhejiang Academy of Medical Sciences, Hangzhou 310012, China; 4School of Public Policy and Management, Tsinghua University, Beijing 100084, China; 5Project Supervision Center of National Health Commission of the People’s Republic of China, Beijing 100044, China; 6Department of Grassroots Public Health Management Group, Public Health Management Branch of Chinese Preventive Medicine Association, Shanghai 201800, China; 7School of Public Health and Management, Chongqing Medical University, Chongqing 400016, China; 8School of Public Health, Shandong University, Jinan 250012, China; 9School of Health Service Management, Anhui Medical University, Hefei 230032, China; 10School of Public Health, Jining Medical University, Jining 272067, China; 11Committee on Medicine and Health of Central Committee of China ZHI GONG PARTY, Beijing 100011, China; 12Institute of Inspection and Supervision, Shanghai Municipal Health Commission, Shanghai 200031, China; 13Shanghai Municipal Health Commission, Shanghai 200031, China; 14Jiangsu Preventive Medicine Association, Nanjing 210009, China; 15Changzhou Center for Disease Control and Prevention, Changzhou 213003, China

**Keywords:** regulatory policy, prevention and control of chronic diseases, community service institution, community health centre, mediation analysis

## Abstract

Regional regulatory policies (RPs) are a major factor in the prevention and control of chronic diseases (PCCDs) through the implementation of various measures. This study aimed to explore the impacts of RPs on PCCDs, with a focus on the mediating roles of community service. The soundness of the regulatory mechanism (SORM) was used to measure the soundness of RPs based on 1095 policy documents (updated as of 2015). Coverage provided by community service institutions (CSIs) and community health centres (CHCs) was used to represent community service coverage derived from the China Statistical Yearbook (2015), while the number of chronic diseases (NCDs) was used to measure the effects of PCCDs based on data taken from the 2015 China Health and Retirement Longitudinal Study survey. To assess the relationship between SORM, NCDs and community service, a negative binomial regression model and mediation analysis with bootstrapping were conducted. Results revealed that there was a negative correlation between SORM and NCDs. CSIs had a major effect on the relationship between RPs and PCCDs, while CHCs had a partial mediating effect. RPs can effectively prevent and control chronic diseases. Increased effort should also be aimed at strengthening the roles of CSIs and CHCs.

## 1. Introduction

The prevalence of chronic noncommunicable diseases constitutes a threat to human life, health, and sustainable development [1]. Nowadays, chronic diseases have become the top risk factors for health, accounting for more than 80% of the 10.3 million annual deaths and 68.6% of the overall economic burden in China [2]. In this context, several factors have seriously threatened people’s health, thus affecting social harmony and the credibility of the Chinese governments, specifically including low awareness rates, long latency, high incidence rates, long clinical courses, high mortality rates, and low control rates [2,3]. These issues demonstrate the need for increased attention on chronic disease, particularly to reduce the risk of illness while improving the overall quality of life.

Chronic diseases are preventable and controllable. Interventions on the prevention and control of chronic diseases (PCCDs) are known to aid in early detection, diagnosis, and treatment initiatives while reducing related economic burdens [4]. Meanwhile, regulatory policies (RPs) can improve the rate of service coverage by promoting the implementation of prevention measures [5], thereby constituting a major source of control for PCCDs. Extensive research has also shown that the soundness of RP has a significant impact on PCCDs [6]. For example, Huang Jianming et al. [7] found that strong community RPs can be used to improve several problematic issues among patients, including hypertension, while Locke et al. [8] demonstrated that RPs had positive effects on the management of diabetic patients in Canada, and Francisco et al. [9] proposed the implementation of multifunctional services designed to monitor and manage COPD patients based on the effectiveness of a chronic disease management model.

The PCCDs is a common and effective practice in many communities [10]. Specifically, in China, community service institutions (CSIs) have become increasingly valuable for PCCDs since the new health care reforms in 2009 [11,12]. CSIs are initiated by the government and mainly responsible for public-welfare-based social service activities; the service scope mainly includes medical care, pensions, and living [13]. Similarly, community health centres (CHCs), which are a smaller unit of CSIs, provide medical and health services and implement important strategies for PCCDs at the community level [14]; this was also observed by Wong [15]. CHCs help to detect risk factors, provide health education, and prevent chronic diseases [16], as has also been reported by Kim and Sharma [17,18]. Therefore, CSIs and CHCs are important factors in promoting health and preventing chronic diseases.

To improve PCCDs, it is necessary to understand the relationship between RPs, CSIs, CHCs, and PCCDs. According to the Donabedian’s model, which contains structure, process, outcome, and the relationship among them (structure-process-outcome) [19], RPs (structure) influence the effects of PCCDs (outcome) through implementation at the process level. CSIs and CHCs, as the ‘gatekeepers’ of community members and ‘vanguards’ of disease prevention and control, provide social support and prevention services, which can effectively control the spread and deterioration of chronic diseases (process) [20]. The two can work together to implement the chronic disease control. We, therefore, assumed that the relationship between RPs and PCCDs could be explained based on both the coverage of CSIs and CHCs. Many studies have adopted mediation analysis. For instance, Vedanthan et al. found that community health workers mediated hypertension care policy and the control of blood pressure [21]. Chen et al. reported that social capital has an effect on physical activity and nutrition [22]. However, no available study in the context of the relationship between RPs and PCCDs and the pathways has adopted mediation analysis. Therefore, this study aimed to fill this gap by exploring the association between RPs and PCCDs, with a focus on whether CSIs and CHCs mediated this relationship.

## 2. Materials and Methods

### 2.1. Study Design

The study setting included 31 provinces in China. A mediation analysis was conducted to determine whether CSIs and CHCs mediated the relationship between RPs and PCCDs. We, therefore, hypothesised the following (Figure 1):

**Hypothesis** **1a** **(H1a).**
*The soundness of RPs has a positive impact on the coverage of CSIs (CHCs).*


**Hypothesis** **1b** **(H1b).**
*The coverage of CSIs (CHCs) has a positive impact on PCCDs.*


**Hypothesis** **1c** **(H1c).**
*The soundness of RPs has a positive impact on PCCDs.*


### 2.2. Measurements

#### 2.2.1. The Soundness of RPs

We constructed the soundness of the regulatory mechanism (SORM) to evaluate the soundness of RPs. The SORM should have at least three characteristics: comprehensiveness, authority, and implementation [23]. Here, the comprehensiveness of the mechanism requires the coverage of essential elements in the policies for PCCDs and clearly outlined responsibilities for each of the departments covered, based on the services they provide [24]. The authority of the mechanism requires issuance of documents by authorities (legislature, government, health commission, etc.) to reflect the importance of RPs. Furthermore, the mechanism requires supervision by external restraint mechanisms for the implementation of the policies [25]. Therefore, four quantitative indicators were adopted. These were named regulatory element coverage rate (RECR), departmental responsibilities clarity rate (DPCR), regulatory mechanism authority rate (RMAR), and accountability mechanism clarity rate (AMCR), and their definitions are detailed in Table 1 [23]. Additionally, SORM was measured based on the sum of the weights of the four quantitative indicators [26]. The study assumed that the higher the SORM (range from 0 to 100 percent), the better the soundness of RPs [26].

#### 2.2.2. PCCDs

The number of chronic diseases (NCDs) was used to evaluate the effects of PCCDs. In this regard, we defined chronic diseases as those with long durations and slow development [27]. A total of 13 chronic diseases were considered according to the standards of data collection presented in the China Health and Retirement Longitudinal Study (CHARLS), including heart disease, stroke, malignant tumour, asthma, chronic obstructive pulmonary disease, diabetes, hypertension, dyslipidaemia (WHO) [28], memory-related diseases (e.g., Alzheimer’s disease, brain atrophy, Parkinson’s disease), kidney disease (excluding tumours and cancers), arthritis/rheumatism, liver disease (excluding fatty liver, tumours, and cancers), and chronic gastroenteritis (ICD-11) [28].

#### 2.2.3. The Coverage of CSIs and CHCs

The coverage of CSIs was measured using the coverage rate of the community service institutions (CRCSI), while the coverage of CHCs was measured using the coverage rate of the community health centres (CRCHC).

### 2.3. Data Collection

Data were collected from the quantitative analysis of policy contents (which contributed to the soundness of RPs), CHARLS (contributed to PCCDs), and China Statistical Yearbook (contributed to CSIs and CHCs).

We first obtained SORM data from policy contents (the coding template is shown in Table A1). Here, we collected a range of policy documents, which were either publicly available or extracted with permission [29] from official websites, including those run by relevant government entities and public health agencies focused on chronic diseases in China (updated as of 2015). The types of policy documents included laws, regulations, plans, guidelines, and others, resulting in a total of 1095 documents from 31 provinces in China.

This study’s researchers were previously trained and thus understood the standardised methods for collecting necessary documents. The coding information mainly included two components, the first of which consisted of basic information related to the documents, including their official names, types, year of publication, and department or institution of publication, while the second consisted of contents related to RPs aimed at chronic disease, including content forms (e.g., long-term goals and short-term goals) [25], responsibilities, work contents, tasks, assessment indicators, assessment requirements, and accountability. We analysed the credibility of the data collection process via the test-retest reliability method with intraclass correlation coefficient (ICC). After two experienced researchers conducted the retest, the ICC was found to be 0.997; as this was greater than 0.75, the data collection process was of high credibility.

Next, NCDs data were drawn from the 2015 CHARLS survey and set as dependent variables. CHARLS was a longitudinal survey that aimed to be representative of the residents in mainland China aged 45 and older, with no upper age limit [30]. Multi-stage stratified probability-proportional-to-size sampling was used to conduct the survey, which randomly selected 150 counties/districts and 450 villages/resident committees, thereby covering 19,000 individuals living in 12,400 households [31] from 28 provinces (data from Hainan, Ningxia, and Tibet were missing), thus collecting a high-quality nationally representative sample [31,32,33,34]. The CHARLS database includes a series of topics, such as demographics, family structure/transfer, health status and functioning, health care and insurance, work, retirement and pension, income and consumption, and community-level information [33]. Subjects with missing or unreasonable responses were excluded from analysis, thus resulting in a final sample size of 16,693.

Finally, CRCSI and CRCHC data were collected from the 2015 China Statistical Yearbook and set as mediating variables.

### 2.4. Statistical Analysis

Data were analysed using both EXCEL 2019 (Microsoft, Redmond, WA, USA) and Stata 14.0 (Stata Corp., College Station, TX, USA).

To avoid an inherent reverse-causality issue, considering that chronic diseases are long duration illnesses [27], we used past RPs to reflect the accumulation of chronic diseases over a long timeframe. The soundness of RPs is a gradual improvement process; therefore, we used the policy contents (updated as of 2015) to comprehensively reflect past regulatory effects. As a result of the accessibility of NCDs, data from 28 provinces were used for descriptive statistics and mediation analysis.

To determine the mediating roles of the CRCSI and CRCHC on the relationship between RPs and PCCDs, we used Spearman’s correlation analysis to assess the multicollinearity between SORM, CRCHC, CRCSI, and NCDs. Multicollinearity meant that highly correlated variables (r > 0.90) or mediation variables that were not correlated with either SORM or NCDs were excluded from the mediation analyses [22].

Subsequently, the following negative binomial regression model was established to analyse the relationship between RPs and PCCDs, as the outcome variable is a count variable with over-dispersed distribution (alpha 95%CI (0.27, 0.31)).
(1)$$\ln\left(\lambda_{i}\right)=\beta_{0}+{\mathrm{cSORM}}_{i}+\delta_{0}X_{i}+\gamma_{0}{\mathrm{Province}}_{i}+\varepsilon_{i}$$

Number of chronic diseases (NCDs) was set as the dependent variable, and $\lambda_{i}$ was the expected count of NCDs. ${\mathrm{SORM}}_{i}$ was set as the independent variable. $X_{i}$ was used as a control variable at the individual level, including age, gender, marital status, education attainment, annual household income per capita, medical insurance, pension insurance, smoking, and drinking. ${\mathrm{Province}}_{i}$ was used as a control variable at the provincial level, including GDP per capita (reflective of the regional economic level), and the proportion of the population over 65 years of age. $\varepsilon_{i}$ were the residuals.

Finally, we conducted a mediation analysis with bootstrapping using 5000 replications [35] and bias-corrected and accelerated confidence interval (BCa CI) [36] to examine whether community service mediated the relationship between SORM and NCD. The mediation method required the following conditions: (1) SORM was significantly associated with NCD (total effect; c coefficient), (2) SORM was significantly associated with CRCSI (CRCHC) (a coefficient), (3) when controlling for SORM, CRCSI (CRCHC) was significantly associated with NCD (b coefficient), (4) the relationship between SORM and NCD was reduced (direct effect, c’ coefficient) when controlling for CRCSI (CRCHC) (indirect effect, a*b). The proportions mediated were determined by dividing the indirect effect (a*b) by the total effect (c coefficient).

## 3. Results

### 3.1. Baseline Characteristics

Table 2 shows the SORM of China and variables at the provincial level in 2015, as used in this study. The SORM was 9.70%, thus indicating substantial room for improvement in regard to the RPs aimed at chronic disease. DPCR was 4.92%, thus indicating that responsibilities should be more clearly defined. The median value of AMCR was 0.55%, thus indicating the lack of an external accountability mechanism in each department, which restricted the effectiveness of PCCDs. The mean of CRCSI (58.46%) was higher than the mean of CRCHC (6.72%).

Table 3 shows baseline characteristics of the study variables. The mean participant age was 61 years, and there was a slightly higher proportion of females (51.85% vs. 48.15% males). A total of 27.36% of participants did not suffer from chronic disease. However, more than 70% had at least one chronic disease, with some suffering from multiple chronic diseases.

### 3.2. Correlation Analysis

Table 4 summarises the relationships between variables using Spearman’s correlation analysis. SORM was positively correlated with both CRCHC (r = 0.331, *p* < 0.01) and CRCSI (r = 0.473, *p* < 0.01). Conversely, NCD exhibited significant negative correlations with SORM (r = −0.029, *p* < 0.01), CRCHC (r = −0.049, *p* < 0.01), and CRCSI (r = −0.059, *p* < 0.01).

### 3.3. Regression Analysis

Table 5 summarises the relationship between SORM and NCDs. SORM exhibited a significant negative correlation with NCD (β = −0.014, *p* < 0.01). When controlling for variables at the provincial level, we found that SORM was still significantly associated with NCD (β = −0.010, *p* < 0.01). In this case, SORM had a positive effect on the NCD.

### 3.4. Mediation Effects

Table 6 shows the mediating effects of CRCHC and CRCSI. There was a negative correlation between SORM and NCD (path: total effect, β = −0.014; 95% CI: −0.022, −0.006). Next, both CRCHC (path: indirect effect, β = −0.002; 95% CI: −0.003, −0.001) and CRCSI (path: indirect effect, β = −0.006; 95% CI: −0.008, −0.005) mediated the relationship between SORM and NCD. However, the bootstrap analysis showed that each played different mediating roles. CRCHC had a partial mediating effect (11.31% of the total), with the 95% CI of the direct effects in model B including the value of zero, while CRCSI had a major mediating effect (45.42% of the total).

## 4. Discussion

This study examined the relationship between RPs and PCCDs, with a focus on the mediating roles of the coverage of CSIs and CHCs. Both policy content and mediation analysis were conducted, demonstrating that RPs had positive impacts on PCCDs, while both the coverage of CSIs and CHCs mediated the relationship between RPs and PCCDs. Further, better SORM was associated with less NCD per capita, which showed that RPs and PCCDs were more effective in regions with those qualities. This supports the relevant literature. For example, previous research has shown that the soundness of RPs can effectively improve unhealthy lifestyles among patients with hypertension [7]. Similarly, the soundness of RPs has also been found to promote maternal and child health [25]. In this context, regions with better RPs have more effective PCCDs, particularly when RPs are more comprehensive and cover a range of aspects, such as diet management and the living environment. Meanwhile, the accountability mechanism is implementable when relevant departments have clear responsibilities. For example, the government may raise taxes on alcohol and tobacco while reducing the available subsidies for unhealthy foods. Environmental protection standards should also be strictly implemented by environmental protection departments. In other areas, the Landscaping bureau is responsible for the construction of urban green spaces, while the Food and Drug Administration must supervise the food processing industry to reduce the use of salt and trans-fatty acids [37]. Especially when combined, these efforts can further prevent and control chronic disease. It is therefore necessary to establish a long-term working mechanism for managing general health and chronic disease. Efforts must be directed at improving the contents and processes of management services, clarifying departmental responsibilities, strengthening the departmental accountability mechanism, and establishing a management model targeted at the integrated prevention, treatment, and management of chronic disease [38].

Our study indicated that CHCs played a partial mediating role in the relationship between RPs and PCCDs. As subsections of CSIs, CHCs were particularly effective entities for controlling chronic disease [39]. Previous studies have shown that communities provide better social environments, which facilitate access to elements such as health education, early detection, early treatment, comprehensive management [40], and rehabilitation exercises [41], all of which can further aid in the control of chronic disease. CHCs play important roles due to important aspects of social mobilisation and policy support [42], as they work as the ‘gatekeeper’ of residents and ‘vanguard’ of prevention and control [20]. Community health personnel also maintain closer relationships with residents, thus providing the health management [43] needed to ensure positive health outcomes. In this regard, China should establish and improve RPs specifically aimed at PCCDs, with a strong focus on the incentive mechanism, thus allowing community health personnel to achieve a balance between income and expenditures on capitation fees. In addition, the government should allocate special chronic disease management funds within the per capita public health funds in order to implement a pay-for-performance provision for community health personnel [44]. This will enhance enthusiasm among community health personnel while continuously strengthening the effects of PCCDs in various regions of China.

We found that CSIs played a major mediating role in the relationship between RPs and PCCDs. The mediation analysis specifically demonstrated that the coverage of CSIs had a more significant mediating effect than that derived from the coverage of CHCs. This may be due to the widespread nature of social factors that affect chronic disease, thus indicating that PCCDs should focus on cooperating with other fields at all levels. As effective carriers for PCCDs, CSIs often undertake multiple responsibilities in addition to those related to health services [10], such as providing social support at community service centres [44], promoting physical exercise for the elderly in designated activity rooms [45], employing community healthcare workers [46], providing medical care through community pension service centres [47], and offering health self-education through reading rooms or at community schools [48]. As a specific example, studies have shown that physical exercise, social interaction, access to care, and community service have positive effects on health of older patients with diabetes [49]. Along with the advantages provided through community-based joint prevention and control, these factors play important roles in the context of PCCDs. Based on the idea that health should be included in all policies, we should consider that effective PCCDs may require coordinated participation from multiple stakeholders [50]. In addition to improving CHCs, we should also pay more attention to how multiple departments work as a joint force in the prevention and reduction of chronic disease.

The results of our policy content analysis showed that SORM requires improvement in China. There are no relevant laws for PCCDs, owing to which the government is unable to ensure adherence. There is also a great deal of room to develop completeness among regulatory elements, including the policymaking process, service provisions, the division of responsibility, and information monitoring and evaluation [24]. In addition, the implementation of PCCDs is restricted by factors such as the lack of organisations or professionals and effective multi-department coordination mechanisms [6]. Further, the responsibilities of relevant departments are not sufficiently clear, thereby resulting in the non-assessment of business objectives and the failure to play a positive role in guiding relevant work. Departments also lack an external supervisory mechanism that could effectively restrict their actions. In this context, the government should work to improve laws and regulations related to the control of chronic disease, enhance the contents of chronic disease management, clarify departmental responsibilities, and improve the regulatory and accountability mechanisms. In sum, these efforts will help prevent and reduce the occurrence of chronic disease.

This study also had some limitations. First, the policy content analysis was primarily aimed at evaluating RPs. In the future, relevant verification information should be collected through additional investigations. Second, mediating roles may not be limited to CSIs and CHCs, as both the quality and type of service may affect the relationship between RPs and PCCDs. Third, the factors affecting PCCDs are not limited to RPs; they also include organisations, resources, and the environment. These factors are interrelated and interact within the health system. Additional research is needed to verify the nature of those relationships.

## 5. Conclusions

This study found that CSIs and CHCs play mediating roles in the relationship between RPs and PCCDs. In this regard, RPs can promote the effects of PCCDs, meaning that improved RPs will promote health while preventing and controlling disease. Moreover, increased attention should be placed on the specific roles played by CSIs and CHCs.

## Figures and Tables

**Figure 1 healthcare-09-01058-f001:**
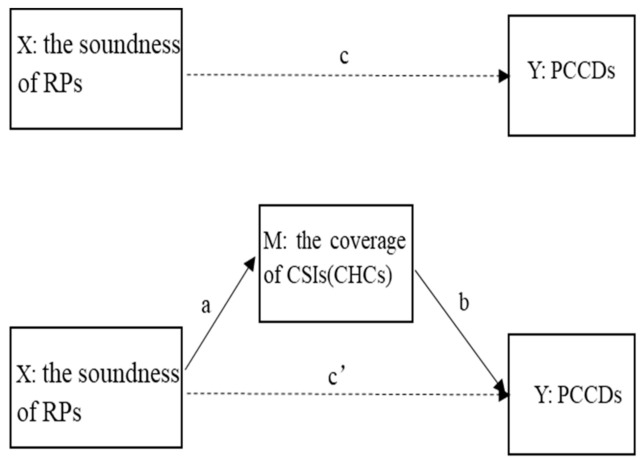
Mediating model for this study’s conceptual framework. Abbreviations: RPs, regulatory policies; CSIs, community service institutions; CHCs, community health centres; PCCDs, prevention and control of chronic diseases.

**Table 1 healthcare-09-01058-t001:** Evaluation indicators for the soundness of RPs.

Characteristics	Quantitative Indicators	Definition of Indicators
Comprehensiveness	RECR (%)	The proportion of the number of regulatory elements covered in a city’s chronic disease health policy document collection to the 25 required elements
	DPCR (%)	The proportion of the number of departments with clear and measurable responsibilities to the 22 departments that should be included in PCCDs
Authority	RMAR (%)	The proportion of the number of authority of government branches and regulatory mechanism document sets to the total required
Implementation	AMCR (%)	The proportion of the number of departments with clearly defined monitoring agencies and accountabilities to the 22 departments that should be included in PCCDs

Abbreviations: RPs, regulatory policies; RECR, regulatory element coverage rate; DPCR, departmental responsibilities clarity rate; RMAR, regulatory mechanism authority rate; AMCR, accountability mechanism clarity rate; PCCDs, prevention and control of chronic diseases.

**Table 2 healthcare-09-01058-t002:** Characteristics of SORM and variables at the provincial level in 2015.

Indicators	Mean Value/Median Value	SD/IQR
SORM (%)	9.70	3.67
RECR (%)	37.84	10.32
DPCR (%)	4.92	4.29
RMAR (%)	23.55	6.91
AMCR (%) ^1^	0.55	(0, 1.85)
Community service		
CRCHC (%)	6.72	6.20
CRCSI (%)	58.46	54.00
Economic and aging level		
GDP per capita (thousand yuan)	54.61	24.00
Proportion of population over 65 years of age (%)	10.35	1.70

^1^ The indicator was expressed as the median (IQR). Abbreviations: SORM, soundness of the regulatory mechanism; RECR, regulatory element coverage rate; DPCR, departmental responsibilities clarity rate; RMAR, regulatory mechanism authority rate; AMCR, accountability mechanism clarity rate; CRCHC, coverage rate of the community health centres; CRCSI, coverage rate of the community service institutions; SD, standard deviation; IQR, interquartile range.

**Table 3 healthcare-09-01058-t003:** Variable descriptions (*n* = 16,693).

Variables	Category	Mean (Median)/n	SD (IQR)/%
NCDs	0	4567	27.36
	1	4481	26.84
	2	3379	20.24
	3	2023	12.12
	4	1162	6.96
	≥5	1081	6.48
Control variables			
Age (years)		61	10.01
Sex	Male = 1	8037	48.15
	Female = 2	8656	51.85
Registered residence	Town = 1	3547	21.25
	Rural = 0	13,146	78.75
Marital status	Married = 1	13,349	79.97
	Other = 0	3344	20.03
Education attainment	Elementary school and below = 1	4900	29.35
	Junior high school = 2	5665	33.94
	Senior high school and above = 3	6128	36.71
Medical insurance	Yes = 1	15,234	91.26
	No = 0	1459	8.74
Pension insurance	Yes = 1	14,696	88.04
	No = 0	1997	11.96
Drinking	Yes = 1	7733	46.33
	No = 0	8960	53.68
Smoking	Yes = 1	6912	41.41
	No = 0	9781	58.59
Annual household income per capita(yuan) ^1^		6993.50	(2470, 17,000)

^1^ The indicator was expressed as the median (IQR). Abbreviations: NCDs, number of chronic diseases; SD, standard deviation; IQR, interquartile range.

**Table 4 healthcare-09-01058-t004:** Spearman’s correlation analysis on the relationships between SORM, CRCHC, CRCSI, and NCD per capita.

Variables	SORM	CRCHC	CRCSI
CRCHC	0.331 ***		
CRCSI	0.473 ***	0.396 ***	
NCD per capita	−0.029 ***	−0.049 ***	−0.059 ***

*** *p* < 0.01. Abbreviations: SORM, soundness of the regulatory mechanism; CRCHC, coverage rate of the community health centres; CRCSI, coverage rate of the community service institutions; NCD, number of chronic diseases.

**Table 5 healthcare-09-01058-t005:** Negative binomial regression analysis of the effects of SORM on NCDs (per capita).

Variables	Model 1		Model 2	
	β	SE	β	SE
SORM	−0.014 ***	0.002	−0.010 ***	0.003
Control variables (individual level)				
Age	0.019 ***	0.001	0.019 ***	0.001
Sex	0.211 ***	0.023	0.213 ***	0.023
Registered residence	0.136 ***	0.019	0.139 ***	(0.019)
Marital status	0.011	0.019	0.011	(0.019)
Education attainment (Reference group: elementary school and below)				
Junior high school	0.082 ***	0.019	0.081 ***	0.019
Senior high school and above	0.028	0.022	0.031	0.022
Log Annual household income per capita	−0.016 ***	0.004	−0.015 ***	0.005
Medical insurance	0.139 ***	0.028	0.136 ***	0.028
Pension insurance	0.049 *	0.025	0.045 *	0.025
Drinking	0.037 **	0.017	0.028 *	0.017
Smoking	0.057 ***	0.021	0.065 ***	0.021
Control variables (provincial level)				
Log GDP per capita			−0.094 ***	0.026
Proportion of population over 65 years of age			0.028 ***	0.005
alpha 95% CI	(0.270, 0.311)		(0.267, 0.308)	

*** *p* < 0.01, ** *p* < 0.05, * *p* < 0.1. Abbreviations: SORM, soundness of the regulatory mechanism; NCDs, number of chronic diseases.

**Table 6 healthcare-09-01058-t006:** Bootstrap analysis of the mediation effects.

Model	Path	β	SE	95% CI	Promotion Mediated
	Total effect	−0.014 ***	0.004	−0.022, −0.006	
A: CRCHC	Direct effect	−0.012 ***	0.004	−0.021, −0.004	11.31%
	Indirect effect	−0.002 ***	0.001	−0.003, −0.001	
B: CRCSI	Direct effect	−0.008 *	0.004	−0.016, 0.001	45.42%
	Indirect effect	−0.006 ***	0.001	−0.008, −0.005	

Bootstrapped standard errors in parentheses *** *p* < 0.01, * *p* < 0.1. We also controlled for age, gender, marital status, education attainment, annual household income per capita, medical insurance, pension insurance, smoking, drinking, GDP per capita, and proportion of the population over 65 years of age. Abbreviations: CRCHC, coverage rate of the community health centres; CRCSI, coverage rate of the community service institutions.

## Data Availability

Some data used in this study are publicly available at http://charls.pku.edu.cn/index/zh-cn.html (accessed on 13 September 2019) and http://www.stats.gov.cn/tjsj/ndsj/2016/indexch.htm (accessed on 26 September 2019). The rest of the data are available from the corresponding author on reasonable request.

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
