# Peer review of "The Impacts of Regional Regulatory Policies on the Prevention and Control of Chronic Diseases in China: A Mediation Analysis"

_healthcare, 2021, doi:10.3390/healthcare9081058_

Round 1
Reviewer 1 Report
Dear Authors,
Your manuscript is very well presented and organised. It provides insightful suggestions for the prevention and control of chronic diseases in China. Please find attached a summary of my suggestion/revisions of the present manuscript. In my opinion, only a very few minor revisions are needed.
Abstract:
I would suggest improving the abstract by adding the background section, which is required. Try to be more concise, since 200 words is the limit that you have already exceeded.
Tables:
Table 2.
Please bear in mind that any tables or figures should be stand-alone, thus comprehensible by themselves all the time. Provide the explanation of each acronym you have used in tables and figures (e.g. SORM, RECR, etc) in the caption. Apply the same to all the presented tables. Moreover, some variable seems quite spread (i.e. AMCR %) as indicated by the wide standard deviation. In these cases, I would suggest presenting the data as median(IQR) instead of mean(SD).
Table 4.
Please indicate in the table caption/description the statistical analysis you have used (i.e. Spearman correlation).
References:
Please double-check references details and be sure to use a consistent formatting throughout (e.g. all years of publication should be bolded, etc).
Best regards
Reviewer 2 Report
This manuscript is focussed on suitability of regional regulatory policies in the prevention of chronic diseases.
I find some problem in the manuscript in the aspect that it is difficult to relate the methods used for determining the soundness of RPs and the survey done with the patients.
What questions were asked to the people enrolled in the survey that could be related with the regulations?
Without this link for me is very difficult to follow the results related to this aspect
Other things:
- Was there any limit age for taking the survey?
- Table 3 last line (Log family income by capita), what units it has?
Round 2
Reviewer 2 Report
The authors have answered satisfactorily, and have made the changes in the text.